# IL-4 Augments IL-31/IL-31 Receptor Alpha Interaction Leading to Enhanced Ccl 17 and Ccl 22 Production in Dendritic Cells: Implications for Atopic Dermatitis

**DOI:** 10.3390/ijms20164053

**Published:** 2019-08-20

**Authors:** Sho Miake, Gaku Tsuji, Masaki Takemura, Akiko Hashimoto-Hachiya, Yen Hai Vu, Masutaka Furue, Takeshi Nakahara

**Affiliations:** 1Department of Dermatology, Graduate School of Medical Sciences, Kyushu University, Maidashi 3-1-1, Higashiku, Fukuoka 812-8582, Japan; 2Research and Clinical Center for Yusho and Dioxin, Kyushu University, Maidashi 3-1-1, Higashiku, Fukuoka 812-8582, Japan; 3Division of Skin Surface Sensing, Graduate School of Medical Sciences, Kyushu University, Maidashi 3-1-1, Higashiku, Fukuoka 812-8582, Japan

**Keywords:** IL-31, IL-31 receptor alpha, Ccl 17, Ccl 22, dendritic cell, atopic dermatitis

## Abstract

Severe pruritus is a characteristic feature of atopic dermatitis (AD) and is closely related to its activity. Recent studies have shown that IL-31 is a key determinant of pruritus in AD. Anti-IL-31 receptor alpha (IL-31RA) antibody treatment has also been reported to improve pruritus clinically, subsequently contributing to the attenuation of AD disease activity. Therefore, IL-31 has been thought to be an important cytokine for regulating pruritus and AD disease activity; however, how IL-31 is involved in the immune response in AD has remained largely unknown. Epidermal Langerhans cells (LCs) and dermal dendritic cells (DCs) derived from bone marrow cells have been reported to play a critical role in AD pathogenesis. LCs and DCs produce Ccl 17 and Ccl 22, which chemoattract Th2 cells, leading to AD development. Therefore, we aimed to clarify how IL-31/IL-31RA interaction affects Ccl 17 and Ccl 22 production. To test this, we analyzed murine bone marrow-derived DCs (BMDCs) stimulated with IL-4, an important cytokine in AD development. We found that IL-31RA expression was upregulated by IL-4 stimulation in a dose-dependent manner in BMDCs. Furthermore, IL-31 upregulates Ccl 17 and Ccl 22 production in the presence of IL-4, whereas IL-31 stimulation alone did not produce Ccl 17 and Ccl 22. These findings suggest that IL-4 mediates IL-31RA expression and IL-31/IL-31RA interaction augments Ccl 17 and Ccl 22 production in BMDCs, which promotes Th2-deviated immune response in AD. Since we previously reported that soybean tar Glyteer, an aryl hydrocarbon receptor (AHR) ligand, impairs IL-4/Stat 6 signaling in BMDCs, we examined whether Glyteer affects IL-31RA expression induced by IL-4 stimulation. Glyteer inhibited upregulation of IL-31RA expression induced by IL-4 stimulation in a dose-dependent manner. Glyteer also inhibited Ccl 17 and Ccl 22 production induced by IL-4 and IL-31 stimulation. Taken together, these findings suggest that Glyteer treatment may improve AD disease activity by impairing IL-31/IL-31RA interaction in DCs.

## 1. Introduction

Interleukin (IL)-31 is a four-helix bundle cytokine closely related to the IL-6 cytokine family [1]. It is the only known ligand for IL-31 receptor alpha (IL-31RA), which belongs to the glycoprotein 130 receptor family [2]. IL-31/IL-31RA interaction initiates signal transduction resulting in the expression of several pro-inflammatory cytokines and chemokines, which play a crucial role in inflammatory diseases relating to Th2 cytokines including IL-4 and IL-13 [3,4].

Atopic dermatitis (AD) is a chronic or chronically relapsing, eczematous, and severely pruritic skin disorder. The lesional skin of AD exhibits Th2-deviated immune reactions such as those involving IL-4, IL-5, and IL-13, which contribute to the development of this disease [5]. It has been shown that the anti-IL-4 receptor alpha (IL-4RA) antibody dupilumab causes marked and rapid improvement in AD patients [6], indicating that IL-4 plays a critical role in AD development. A recent study has also shown that anti-IL-31 receptor alpha (IL-31RA) antibody treatment using nemolizumab produced clinical improvement of pruritus, subsequently contributing to the attenuation of AD disease activity [7]. The mechanism behind this is still not fully understood, but it has been shown that IL-31RA is expressed on sensory neurons and IL-31 derived from activated T cells under Th2-skewed conditions stimulates IL-31RA on sensory nerves, resulting in severe pruritus in AD [8]. Although IL-31/IL-31RA interaction has been identified as one of the key determinants of pruritus in AD, there are limited reports regarding the effect of IL-31/IL-31RA interaction on Th2-deviated immune conditions in AD [4,9]. It has been reported that transgenic mice overexpressing IL-31 develop AD-like skin lesions with severe pruritus [10]. In addition, the neutralization of IL-31 and genetic inhibition of EPAS1, a regulator of IL-31 induction in CD4+ T cells, have been reported to ameliorate AD-like skin lesions [11]. Therefore, IL-31/IL-31RA interaction is deeply involved in AD disease activity in addition to pruritus. 

Given that (1) CCL 17 and CCL 22 chemoattract Th2 cells and maintain the Th2 immune response [12,13], (2) serum CCL 17 and CCL 22 levels are closely related to the disease activity of AD [14], and (3) Langerhans cells and dermal dendritic cells (DCs) produce prominent CCL 17 and CCL 22 in the skin of AD patients [12,15], we investigated whether IL-31/IL-31RA interaction affects CCL 17 and CCL 22 production in DCs during the development of AD. To test this, we analyzed murine bone marrow-derived DCs (BMDCs) treated with IL-4, which is thought to recapitulate the conditions of skin myeloid DCs in AD. 

## 2. Results

### 2.1. IL-4 Stimulation Upregulated IL-31RA Expression in BMDCs

We first examined whether IL-4 stimulation upregulated IL-31RA expression in BMDCs. We analyzed BMDCs stimulated with IL-4 (0.1, 1, and 10 ng/mL) for 24 h. Quantitative reverse-transcription (qRT-PCR) analysis revealed that IL-4 stimulation upregulated IL-31RA expression at the mRNA level in a dose-dependent manner (Figure 1a). Flow cytometry analysis of BMDCs stimulated with IL-4 (10 ng/mL) for 24 h also showed that IL-4 stimulation upregulated IL-31RA expression at the protein level (Figure 1b). We also examined expression of OSMRβ, the other subunit of IL-31 receptor; however, it was not upregulated by IL-4 stimulation at the mRNA level (Figure 1c) in BMDCs stimulated with IL-4 (0.1, 1 and 10 ng/mL) for 24 h (Figure 1c).

### 2.2. IL-31 Stimulation Enhanced IL-4-Induced Ccl 17 and Ccl 22 Production in BMDCs

In addition to our previous report [15] describing that IL-4 stimulation induced Ccl 17 and Ccl 22 production in BMDCs, we next examined how IL-31 stimulation affects Ccl 17 and Ccl 22 production. We analyzed Ccl 17 and Ccl 22 expression in BMDCs stimulated with or without IL-4 (10 ng/mL) and IL-31 (50 and 100 ng/mL) for 24 h. qRT-PCR analysis showed that IL-31 stimulation alone (50 and 100 ng/mL) did not induce upregulation of Ccl 17 and Ccl 22 mRNA expression. Alternatively, IL-31 (50 and 100 ng/mL) with IL-4 (10 ng/mL) stimulation enhanced expression of Ccl 17 and Ccl 22 rather than IL-4 alone at the mRNA level (Figure 2a,c). ELISA analysis of the culture medium of BMDCs stimulated with IL-31 (50 and 100 ng/mL) with IL-4 (10 ng/mL) for 48 h also showed that IL-31 (50 and 100 ng/mL) with IL-4 (10 ng/mL) stimulation enhanced production of Ccl 17 and Ccl 22 rather than IL-4 alone at the protein level (Figure 2b,d). These results suggest that IL-4-induced IL-31RA upregulation is functional through IL-31/IL-31RA interaction, which leads to the enhanced production of Ccl 17 and Ccl 22 in BMDCs. Since another previous report stimulation with a high concentration (250 ng/mL) of IL-31 led to CCL22 production in human monocyte-derived DCs [16], we examined whether IL-31 stimulation (250 ng/mL) affects Ccl 22 expression. In our study, IL-31 stimulation alone (250 ng/mL) did not induce upregulation of Ccl 22 expression at the mRNA level (Appendix A) and the protein level (Appendix A), in BMDCs.

### 2.3. Glyteer Treatment Inhibited IL-4-Induced IL-31RA Expression in BMDCs

Since we previously reported that Glyteer treatment impairs the IL-4/STAT6 signaling pathway in BMDCs [15] and human keratinocytes [17], we examined whether Glyteer treatment also inhibits IL-31RA expression induced by IL-4 stimulation in BMDCs. We analyzed BMDCs stimulated with IL-4 (10 ng/mL) for 24 h in the presence or absence of Glyteer (10^−5^, 10^−6^, and 10^−7^%). qRT-PCR analysis showed that Glyteer treatment inhibited IL-4-induced IL-31RA expression at the mRNA level in a dose-dependent manner (Figure 3a). Flow cytometry analysis of BMDCs stimulated with IL-4 (10 ng/mL) for 24 h in the presence or absence of Glyteer (10^−5^%) also showed that IL-31RA-positive cells increased upon stimulation with IL-4 (10 ng/mL), compared with that in the control group, which was inhibited by Glyteer treatment (10^−5^%) (around the area indicated by a white arrow).

### 2.4. Glyteer Treatment Inhibited IL-4- and IL-31-Induced Ccl 17 and Ccl 22 Production in BMDCs

To further confirm the inhibitory effects of Glyteer treatment, we examined whether the enhancement of Ccl 17 and Ccl 22 production by IL-31 with IL-4 stimulation was inhibited in BMDCs. We analyzed Ccl 17 and Ccl 22 expression in BMDCs stimulated with or without IL-4 (10 ng/mL) and IL-31 (50 and 100 ng/mL) for 24 h in the presence or absence of Glyteer (10^−5^, 10^−6^ and 10^−7^%). qRT-PCR analysis showed that Glyteer treatment inhibited the upregulation of Ccl 17 and Ccl 22 induced by IL-4 stimulation at the mRNA level (Figure 4a,c). ELISA analysis of the culture medium of BMDCs stimulated with or without IL-4 (10 ng/mL) and IL-31 (50 and 100 ng/mL) for 48 h in the presence or absence of Glyteer (10^−5^, 10^−6^ and 10^−7^%) also showed that Glyteer treatment inhibited the upregulation of Ccl 17 and Ccl 22 induced by IL-4 stimulation at the protein level (Figure 4b,d). These results are already proven in our previous report [15]. Furthermore, Glyteer treatment inhibited the upregulation of Ccl 17 and Ccl 22 by IL-31 with IL-4 stimulation at the mRNA level (Figure 4a,c) and the protein level (Figure 4b,d) in a dose-dependent manner. 

## 3. Discussion

Although it has been reported that the IFN-γ/STAT1 signal mediates IL-31RA expression in DCs [16], whether Th2 cytokines such as IL-4 and IL-13 can upregulate IL-31RA expression in DCs has not been examined. Since recent studies have shown that IL-31 and its interaction with IL-31RA play a crucial role in the development of AD [7,11], clarifying the regulatory mechanism of IL-31RA expression under Th2-deviated conditions is very useful for treating AD. The present study has shown for the first time that IL-4 stimulation is capable of upregulating IL-31RA in BMDCs, which is consistent with a previous report describing that either IL-4 or IL-13 can upregulate IL-31RA via Stat6 in murine macrophages [4].

To further determine whether the IL-4-induced IL-31RA is functional, we stimulated IL-4-treated BMDCs with IL-31. This IL-31 stimulation without IL-4 treatment did not induce Ccl 17 and Ccl 22 production, indicating that the upregulation of IL-31RA expression is required for Ccl 17 and Ccl 22 production induced by IL-31, which is partially consistent with a previous report describing that CCL 17 is not produced by IL-31-stimulated human DCs [16]. Nevertheless, the IL-31 stimulation with IL-4 treatment enhanced Ccl 17 and Ccl 22 production compared with IL-4 treatment alone. These findings suggest that a Th2-prone immune condition including IL-4 treatment augments IL-31/IL-31RA interaction, which enhances Ccl 17 and Ccl 22 production and subsequently exacerbates the disease activity of AD. 

The mechanism by which IL-31 stimulation enhances IL-4-induced Ccl 17 and Ccl 22 production remains incompletely understood. Several studies have reported that IL-31/IL-31RA interaction mediates STAT1/3/5 and MAP kinase activation [18,19]. Based on the evidence (1) that STAT1 activation induces CCL 17 and CCL 22 in HaCaT cells (human keratinocytes) [20], and (2) that the inhibition of MAP kinase activation reduces CCL 17 and CCL 22 production in HaCaT cells [21], multiple pathway signal transduction such as STAT1/3 and MAP kinase induced by the IL-31/IL-31RA interaction may be involved in the enhancement of Ccl 17 and Ccl 22 production by IL-31 stimulation. However, further studies are required to confirm this.

Since Glyteer inhibits IL-4-induced upregulation of IL-31 RA expression and IL-4-induced Ccl 17 and Ccl 22 production in BMDCs, the dual effects of Glyteer on IL-31RA expression and Ccl 17 and Ccl 22 production contribute to its inhibition of Ccl 17 and Ccl 22 production in AD. These findings suggest that Glyteer treatment potentiates the efficiency of nemolizumab, an IL-31 receptor α-antagonist, and dupilumab, an IL-4 receptor α-antagonist, in the treatment of AD. Nemolizumab has recently been used for the treatment of AD patients and shown to significantly improve clinical outcomes [7]. Although IL-31 is considered to be an important cytokine in the pruritus of AD, the present study revealed a new role of IL-31 in Th2-deviated immune conditions such as the production of CCL 17 and CCL 22 in AD.

In conclusion, we have demonstrated (i) that IL-31/IL-31RA interaction in dendritic cells under Th2-skewed conditions may increase the production of CCL17 and CCL22, contributing to the disease activity of AD, and (ii) that Glyteer treatment could have potential to prevent the production of CCL17 and CCL22 in AD (Figure 5). These findings should also promote our understanding of the roles of dendritic cell functions in AD.

## 4. Materials and Methods

### 4.1. Reagents

DMSO was purchased from Sigma-Aldrich (St. Louis, MO, USA). Recombinant murine IL-31 and IL-4 were purchased from PeproTech (Rocky Hill, NJ, USA). Glyteer was provided by Fujinaga Pharm Co. Ltd. (Tokyo, Japan) as original sticky-liquid stock. Glyteer is generated from soybean as dry distillated tar after delipidation, which contains wide-ranged organic compounds such as polycyclic aromatic hydrocarbons. Since Glyteer is known to act as a potent AHR ligand, no farther investigations have been conducted to clarify the detailed bioactive components.

### 4.2. Generation of Bone Marrow-Derived DCs (BMDCs) and Cell Culture

C57BL/6J mice were housed and bred in a clean breeding facility, and the trial was approved by the animal facility of Kyushu University (A30-258-0, 20 Aug 2018–31 Mar 2020). Bone marrow cells were collected freshly from femoral and tibial bones of mice and cultured in RPMI 1640 medium (Sigma-Aldrich) containing 1 mmol/L sodium pyruvate (10 mL; Invitrogen, Waltham, MA, USA), 10 mmol/L 4-(2-hydroxyethyl)-1-piperazineethanesulfonic acid (HEPES) (10 mL; Invitrogen, Waltham, MA, USA), 1% Minimum Essential Medium Non-Essential Amino Acids (MEM NEAA) (10 mL; Invitrogen), 10% FBS (Japan Bio Serum, Fukuyama, Japan), β-mercaptoethanol (50 nmol/L; Invitrogen), 100 U/mL penicillin, and antibiotic-antimycotic 100× (5 mL; 100 mg/mL streptomycin, and 0.25 μg/mL amphotericin B; Invitrogen) with GM-CSF (10 ng/mL) (Miltenyi Biotec, Bergisch Gladbach, Germany). Medium was refreshed twice in 7 days. Non-adherent cells were collected on day 7, and cells were purified immunomagnetically by positive selection with CD11c (N418) MicroBeads (Miltenyi Biotec). Our previous study has reported that the purity of BMDCs was 95–97%, that was confirmed by flow cytometric analysis [15]. Purified BMDCs were cultured with/without stimulants such as IL-4 and IL-31, and/or Glyteer for 24 h or 48 h. Culture supernatants were collected for ELISA analysis. Cell pellets were utilized for qRT-PCR or FACS analysis. This method of generating BMDCs and cell culture matches that in our previous study [15].

### 4.3. Quantitative Reverse-Transcription (qRT)-PCR Analysis

Total RNA was extracted by using RNeasy^®^ Mini Kit (Qiagen, Hilden, Germany), then reverse transcription for making cDNA was conducted by using PrimeScript^TM^ RT reagent kit (Takara Bio, Kusatsu, Japan). qRT-PCR was performed on CFX Connect Real-time PCR Detection System (Bio-Rad, Hercules, CA, USA) using TB Green^®^ Premix Ex Taq (Takara Bio, Kusatsu, Japan). Amplification was launched as the first step at 95 °C for 30 s, then followed by 40 cycles of qRT-PCR at 95 °C for 5 s and at 61 °C for 20 s as the second step. mRNA expression was measured with normalization by housekeeping gene using β-actin in triplicate, and the mRNA expressions were presented as the fold induction relative to the control group. Each primer sequences are shown in Table 1.

### 4.4. ELISA

Murine Ccl 17 and Ccl 22 ELISA Kit (R&D Systems, Minneapolis, MN, USA) was utilized to measure production of Ccl 17 and Ccl 22 following the manufacturer’s protocol. Absorbance of the wells of the ELISA plate was measured using DTX 800 Multimode Detector (Beckman Coulter).

### 4.5. Fluorescence-Activated Cell Sorting (FACS) Analysis

LIVE/DEAD fixable dead cell staining kit (violet and near IR; Life Technologies) was used to detect dead cells. Then, BMDCs were incubated with anti-CD16/32 antibody (Becton Dickinson, Franklin Lakes, NJ, USA) for 30 min on ice. Anti-murine IL-31 RA antibody (Bioss Antibodies, Boston, MA, USA) was labeled with Zenon R-Phycoerythrin (PE) Rabbit IgG Labeling Kit (Thermo Fisher Scientific, Waltham, MA, USA), in accordance with the manufacturer’s protocol. IL-31RA was stained with PE-conjugated anti-murine IL-31RA antibody. Isotype-matched antibodies were used as controls. Stained cells were analyzed on a BD Canto flow cytometer (Becton Dickinson). Stained cells were also observed using a D-Eclipse confocal laser scanning microscope (Nikon, Tokyo, Japan).

### 4.6. Statistical Analysis

Two-sample Student’s *t* test was used for statistical evaluation of the results. A *p*-value of <0.05 was considered to show a statistically significant difference. All data are presented as mean ± standard error of the mean (S.E.M.) from three independent experiments.

## Figures and Tables

**Figure 1 ijms-20-04053-f001:**
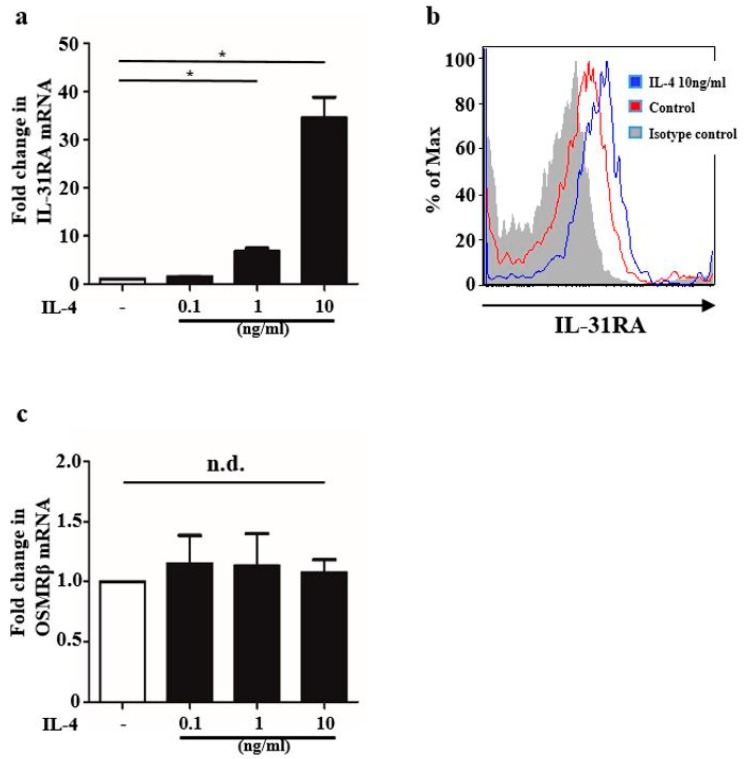
IL-4 stimulation upregulated IL-31RA expression in bone marrow-derived dendritic cells (BMDCs). (**a**,**c**) Data are expressed as mean ± standard error of the mean (S.E.M.); *n* = 3 for each group; * *p* < 0.05. Expression of IL-31RA (**a**) and OSMRβ (**c**) mRNA in BMDCs stimulated with IL-4 (0.1, 1 and 10 ng/mL) for 24 h. mRNA levels normalized for ACTB are expressed as fold induction compared with that in the control group. ACTB was utilized as a housekeeping gene. (**b**) BMDCs were treated with or without IL-4 (10 ng/mL) for 24 h. IL-31RA expression was evaluated using anti-murine IL-31RA antibody. Data are representative of experiments repeated three times with similar results.

**Figure 2 ijms-20-04053-f002:**
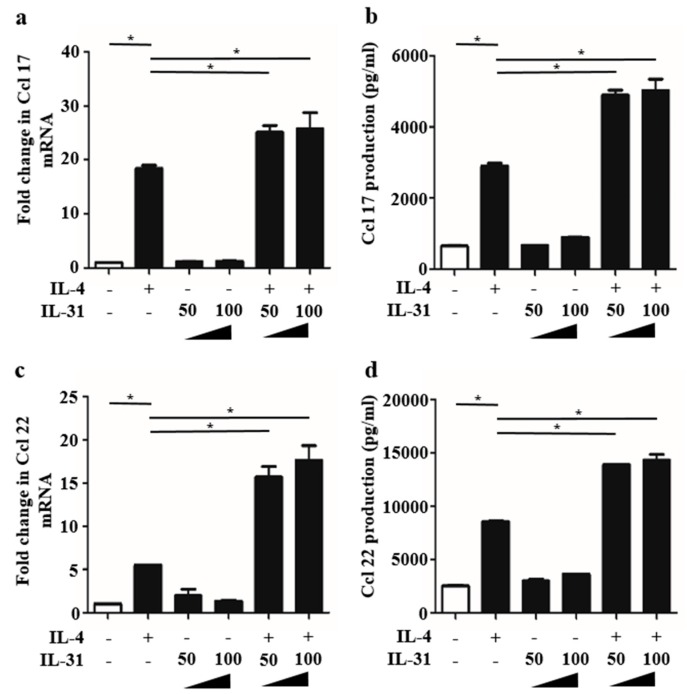
IL-31 stimulation enhanced IL-4-induced Ccl 17 and Ccl 22 production in BMDCs. (**a**–**d**) Data are expressed as mean ± standard error of the mean (S.E.M.); *n* = 3 for each group; * *p* < 0.05. Expression of Ccl 17 (a) and Ccl 22 (c) mRNA in BMDCs stimulated with or without IL-4 (10 ng/mL) and IL-31 (50 and 100 ng/mL) for 48 h and production of Ccl 17 (b) and Ccl 22 (d) in the culture supernatant were measured.

**Figure 3 ijms-20-04053-f003:**
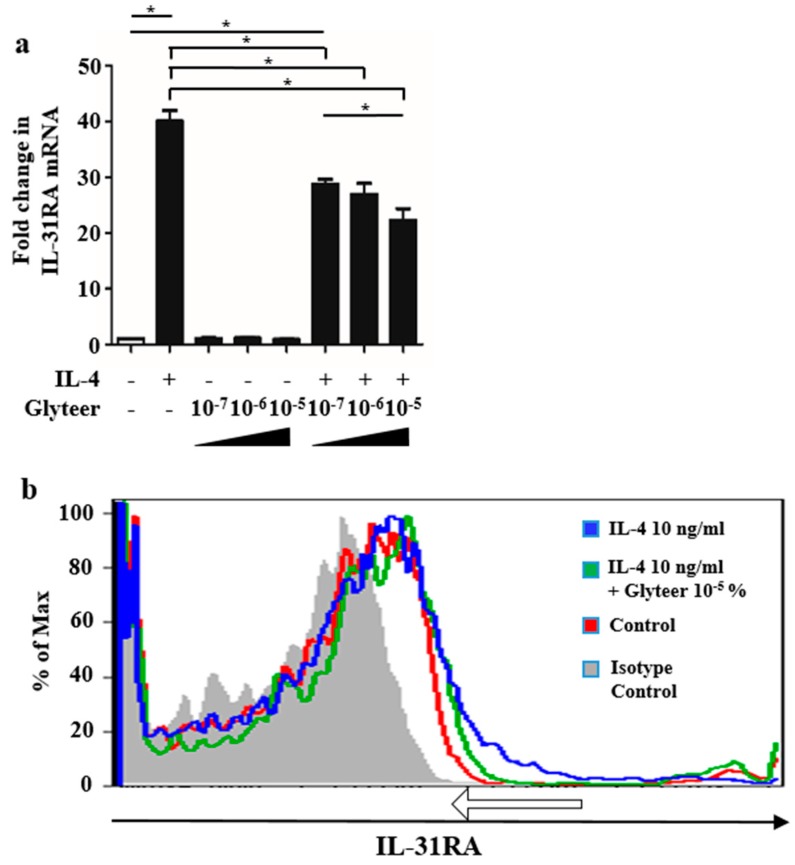
Glyteer impaired IL-4-induced IL-31RA expression. (**a**) Data are expressed as mean ± standard error of the mean (S.E.M.); *n* = 3 for each group; * *p* < 0.05. Expression of IL-31RA mRNA in BMDCs stimulated with or without IL-4 (10 ng/mL) and Glyteer (10^−7^, 10^−6^ and 10^−5^%) for 24 h. (**b**) BMDCs were treated with or without IL-4 (10 ng/mL) and Glyteer (10^−5^%) for 24 h. IL-31RA expression was evaluated using anti-murine IL-31RA antibody. IL-31RA-positive cells increased upon stimulation with IL-4 (10 ng/mL), compared with that in the control group, which was inhibited by Glyteer treatment (10–5%) (around the area indicated by a white arrow). Data is representative of experiments repeated three times with similar results.

**Figure 4 ijms-20-04053-f004:**
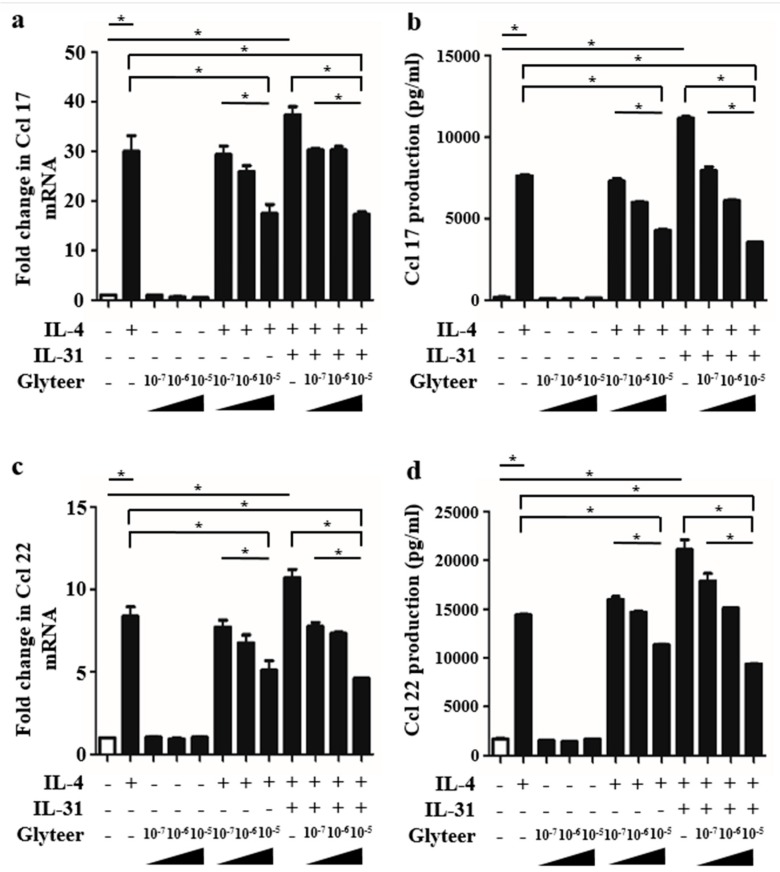
Glyteer impaired IL-4 with IL-31-induced Ccl 17 and Ccl 22 production in BMDCs. (**a**–**d**) Data are expressed as mean ± standard error of the mean (S.E.M.); *n* = 3 for each group; * *p* < 0.05. Expression of Ccl 17 (**a**) and Ccl 22 (**c**) mRNA in BMDCs stimulated with or without IL-4 (10 ng/mL), IL-31 (100 ng/mL), and Glyteer (10^−7^, 10^−6^ and 10^−5^%) for 48 h and production of Ccl 17 (**b**) and Ccl 22 (**d**) in the culture supernatant were measured.

**Figure 5 ijms-20-04053-f005:**
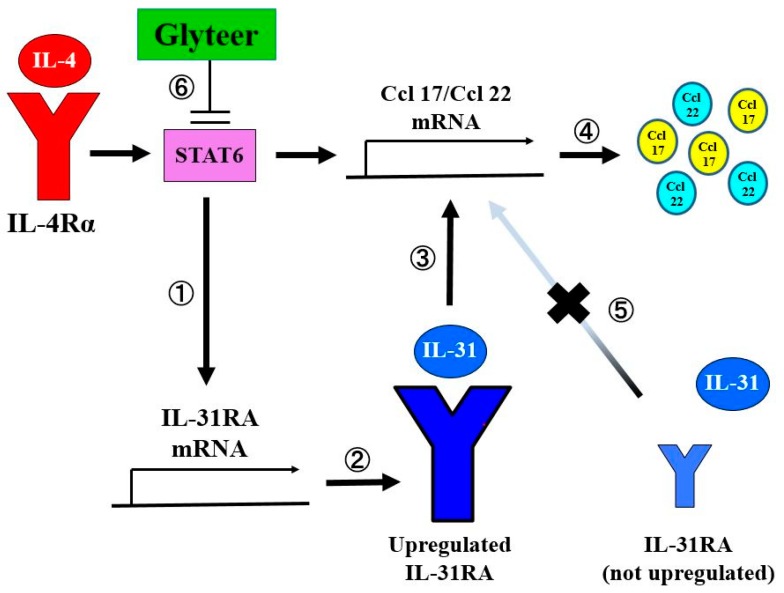
IL-4 augments IL-31/IL-31 RA interaction leading to enhanced Ccl 17 and Ccl 22 production in BMDCs. IL-4 induces upregulation of IL-31RA expression at the mRNA levels ① and the protein levels ②. IL-31RA expression initially upregulated by IL-4 stimulation can lead upregulation of Ccl 17 and Ccl 22 production at the mRNA levels ③ and the protein levels ④, whereas IL-31 stimulation alone without IL-31RA upregulation did not upregulate Ccl 17 and Ccl 22 production ⑤. Impairment of STAT6 signaling pathway activation by Glyteer ⑥ inhibits upregulation of IL-31RA expression and Ccl 17 and Ccl 22 production induced by IL-4 in BMDCs.

**Table 1 ijms-20-04053-t001:** The sequences of primers.

Gene		Sequence (5’ to 3’)
*β-actin*	forward	GGCTGTATTCCCCTCCATCG
	reverse	CCAGTTGGTAACAATGCCATGT
*Ccl17*	forward	AGGTCACTTCAGATGCTGCTC
	reverse	ACTCTCGGCCTACATTGGTG
*Ccl22*	forward	GACACCTGACGAGGACACA
	reverse	GCAGAGGGTGACGGATGTAG
*IL-31RA*	forward	CGATTGTTGTGGAAGAAGGCAA
	reverse	TACTGCTGGGTGGTGATGTTG
*OSMRβ*	forward	CAGGCGGGTAATCAGACCAATG
	reverse	CATGAGTAAGGGCTGGGACA

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
