# Peer review of "IL-4 Augments IL-31/IL-31 Receptor Alpha Interaction Leading to Enhanced Ccl 17 and Ccl 22 Production in Dendritic Cells: Implications for Atopic Dermatitis"

_ijms, 2019, doi:10.3390/ijms20164053_

Round 1

Reviewer 1 Report

The paper describes the mechanism through which IL-31/IL23 31RA interaction affects Ccl 17 and Ccl 22 production as well as the implications in atopic dermatitis. The authors have conducted a rigorous scientific work and produced an overall well written manuscript. The experiments were carefully planned and executed, the results clearly presented and discussed according to previously published literature. I would have the following recommendation:

- in the introduction section, the last paragraph (lines 68-78) contains in my opinion too many informations regarding the experimental results. I would recommend the authors to put those information in the Results section and to leave only the aim of their study in the introduction. 

- in the Discussion section (given the fact that the paper does not have a separate Conclusion section), the authors should introduce a final paragraph to present their own overall conclusion of their study, based on previously analysed results.

Author Response

- in the introduction section, the last paragraph (lines 68-78) contains in my opinion too many information regarding the experimental results. I would recommend the authors to put those information in the Results section and to leave only the aim of their study in the introduction. 

→ Thank you very much for your comment. As you recommended, we modified the style of the manuscript. In detail, we moved parts of the Introduction to the Results section and amended them.

- in the Discussion section (given the fact that the paper does not have a separate Conclusion section), the authors should introduce a final paragraph to present their own overall conclusion of their study, based on previously analyzed results.

→ Thank you very much for your comment.

We amended the parts of the Discussion by adding a last paragraph as follows:

 “In conclusion, we have demonstrated i) that IL-31/IL-31RA interaction in dendritic cells under Th2-skewed conditions may increase the production of CCL17 and CCL22, contributing to the disease activity of AD, and ii) that Glyteer treatment could have potential to prevent the production of CCL17 and CCL22 in AD. These findings should also promote our understanding of the roles of dendritic cell functions in AD.”

 (page 8, lines 206–210)

Reviewer 2 Report

This manuscript demonstrates the role of IL-4 in accelerating IL-31/IL-31R function to induce Cccl17 and Ccl22 in dendritic cells associated with atopic dermatitis. This is an interesting article but it’s an extension of their previous publication (Takemura et al, 2018).

There are several limitations of this manuscript-

1.     Page 2, line 59: Authors mentioned that “there is possibility that IL-31/IL-31RA interaction is deeply involved in AD….”. IL-31/IL-31RA interaction is already known for AD, not sure why authors used “possibility”.

2.     Page 3, Figure 1(c), images don’t look good, it’s really hard to interpret that these are membrane staining, they could use higher magnification.

3.     Page 4, fig 1(d): OSMRb expression is decreased at 0.1 ng/ml IL-4 treatment and it looks it’s significant, please explain.

4.     Page 5, figure 2 (a, b): IL-4/IL-31 treatment increases Ccl17 expression only about 1.25-fold (~25/20) compared to IL-4 (1a), is it really a synergistic effect? The same treatment could vary with this extent in the different set of experiment. How many sets of the experiment they perform to verify their data?

Their previous article showed Ccl17 protein expression >15000 pg/ml after IL-4 treatment (fig 1b; Takemura at al., 2018) whereas result here (2b) is showing only ~3000 pg/ml with IL-4 which is even lower than the basal level (~5000 pg/ml, fig 1b; Takemura et al., 2018) of their previous data and IL-4/IL-31 treatment shows only ~5000 pg/ml, please explain.

5.     Page 5, figure 2(a): IL-31 treatment (50 and 100 ng/ml) doesn’t show any significant increase in Ccl22 expression, did they try at higher concentration? Hoeck-Horejs et al., 2012 showed a significant increase in CCL22 expression in human DCs after 250ng/ml IL-31 treatment. So, IL-31 itself is capable of inducing CCL22 without IL-4.

6.     Page 6, figure 3 (b): It’s really hard to interpret anything from this figure. Also, most of the results section are extremely short, they should interpret their data quite clearly in the results section.

7.     In all the figures, they used only one p-value (*<0.05), did they get only <0.05 not <01, <0.001?

8.     In the materials and method section, they copied many sections from their previous paper (Takemura et al., 2018) but they didn’t cite that article here. They should add a brief method and cite the previous paper.

 9.     Table 1, most of the primer sequences are completely different from Takemura et al., 2018 paper, any reason?

Author Response

Page 2, line 59: Authors mentioned that “there is possibility that IL-31/IL-31RA interaction is deeply involved in AD….”. IL-31/IL-31RA interaction is already known for AD, not sure why authors used “possibility”.

→ Thank you very much for your comment. We have amended the part as follows:

‘’Therefore, there is a possibility that IL-31/IL-31RA interaction is deeply involved in AD disease activity in addition to pruritus.’’

(Page 2, lines 61-62)

Page 3, Figure 1(c), images don’t look good, it’s really hard to interpret that these are membrane staining, they could use higher magnification.

→ Unfortunately, we were not able to achieve better pictures for Figure 1(c). Since we did not prepare a cut section of BMDCs stained with anti-IL-31RA antibody, it is difficult to determine the localization of IL-31RA expression on the cell membrane. Since we just wanted to make sure that the anti-IL-31RA antibody worked well (it is a different antibody from that used in the work by Edukulla R et al. [J Biol Chem. 2015 May 22;290(21):13510-20], we performed the experiment. Thus, we would like to remove the results in this article. We believe that the results of FACS analysis [Figure 1(b)] are comprehensive for evaluating the upregulation of IL-4-induced IL-31RA expression at the protein level.

Page 4, fig 1(d): OSMRb expression is decreased at 0.1 ng/ml IL-4 treatment and it looks it’s significant, please explain.

 → We evaluated OSMRβ expression again and confirmed that 0.1 ng/ml IL-4 treatment did not alter it. We would like to replace the figure with new Figure 1(c), which shows no difference of OSMRβ expression between the control group and groups treated with different concentrations of IL-4. And we added a sentence as below;

“We also examined expression of OSMRβ, the other subunit of IL-31 receptor; however, it was not upregulated by IL-4 stimulation at the mRNA level (Figure 1c) in BMDCs stimulated with IL-4 (0.1, 1 and 10 ng/ml) for 24 h (Figure 1c)”. (page 2, line 77-79)

Page 5, figure 2 (a, b): IL-4/IL-31 treatment increases Ccl17 expression only about 1.25-fold (~25/20) compared to IL-4 (1a), is it really a synergistic effect? The same treatment could vary with this extent in the different set of experiment. How many sets of the experiment they perform to verify their data?

→ We performed the experiments three times under the same conditions to archive the results using qRT-PCR and ELISA. Since IL-31 stimulation alone did not alter Ccl17 production in BMDCs, we drew the conclusion that the upregulation of Ccl17 production by IL-4 plus IL-31 stimulation involved a synergistic effect.

Their previous article showed Ccl17 protein expression >15000 pg/ml after IL-4 treatment (fig 1b; Takemura at al., 2018) whereas result here (2b) is showing only ~3000 pg/ml with IL-4 which is even lower than the basal level (~5000 pg/ml, fig 1b; Takemura et al., 2018) of their previous data and IL-4/IL-31 treatment shows only ~5000 pg/ml, please explain.

→ Thank you very much for this important comment. As you pointed out, our result [Figure 2(b)] that the Ccl17 protein level peaked at 5,000 pg/ml with IL-4/IL-31 stimulation was lower than the level in our previous report. Since it is difficult to reproduce the same conditions in experiments using BMDCs, we evaluated the data in the present study in comparison with our previous studies. We confirmed that Ccl17 production in the IL-4-stimulated BMDCs was about three times that in the control, which was the same trend as in our previous report (Takemura at al., 2018).  

Page 5, figure 2(a): IL-31 treatment (50 and 100 ng/ml) doesn’t show any significant increase in Ccl22 expression, did they try at higher concentration? Hoeck-Horejs et al., 2012 showed a significant increase in CCL22 expression in human DCs after 250ng/ml IL-31 treatment. So, IL-31 itself is capable of inducing CCL22 without IL-4.

 → We also examined IL-31 stimulation (250 ng/ml) in BMDCs, as you suggested; however, IL-31 stimulation alone (250 ng/ml) did not alter Ccl22 upregulation of mRNA expression and protein production in BMDCs. We presented the data in the supplementary Figure 1, which showed no difference of Ccl 22 production among the different IL-31 concentrations (50, 100, and 250 ng/ml) at the mRNA level [Supplementary Figure 1a] and the protein level [Supplementary Figure 1b]. And we added a sentence as below;

“Since another previous report stimulation with a high concentration (250 ng/ml) of IL-31 led to CCL22 production in human monocyte-derived DCs [18], we examined whether IL-31 stimulation (250 ng/ml) affects Ccl 22 expression. In our study, IL-31 stimulation alone (250 ng/ml) did not induce upregulation of Ccl 22 expression at the mRNA level (Supplemental Figure 1a) and the protein level (Supplemental Figure 1b), in BMDCs.” (page 3, line 99-104)

Page 6, figure 3 (b): It’s really hard to interpret anything from this figure. Also, most of the results section are extremely short, they should interpret their data quite clearly in the results section.

 → Thank you very much for this important comment. We amended the part and the figure, including widening it (Figure 3b). We also added a white arrow that indicates inhibitory effect of Glyteer treatment on the upregulation of IL-31RA expression induced by IL-4 stimulation. We also added detailed descriptions in the Results section as below;

“Flow cytometry analysis of BMDCs stimulated with IL-4 (10 ng/ml) for 24 h in the presence or absence of Glyteer (10-5 %) also showed that IL-31RA-positive cells increased upon stimulation with IL-4 (10 ng/ml), compared with that in the control group, which was inhibited by Glyteer treatment (10-5 %) (around the area indicated by a white arrow).”(page3, line 111-115)

In all the figures, they used only one p-value (*<0.05), did they get only <0.05 not <0.01, <0.001?

→ We re-evaluated the p-values and confirmed that no values of <0.01 were obtained.

In the materials and method section, they copied many sections from their previous paper (Takemura et al., 2018) but they didn’t cite that article here. They should add a brief method and cite the previous paper.

→ As you pointed out, this is a continuation of our previous paper. We added new descriptions in the Materials and Methods in the present study and cited the previous paper. We added a sentence as below;

“This method of generating BMDCs and cell culture matches that in our previous study [16]” (page 8, line 322)

Table 1, most of the primer sequences are completely different from Takemura et al., 2018 paper, any reason?

→ We double-checked the details of the primer sequences. We apologize for the mistakes of the descriptions of the primer sequences of β-actin, Ccl17, and Ccl22. Those three primers are the same as used in our previous study. We also added the primer sequences of OSMRβ. We amended the relevant part of Table 1.

Round 2

Reviewer 2 Report

Authors responses are satisfactory.